# Electrochemical Characterization and Voltammetric Determination of Methylisothiazolinone on a Boron-Doped Diamond Electrode

**DOI:** 10.3390/molecules27249013

**Published:** 2022-12-17

**Authors:** Magdalena Jakubczyk, Slawomir Michalkiewicz, Agata Skorupa, Kinga Krajcarz

**Affiliations:** 1Institute of Chemistry, Jan Kochanowski University, PL-25406 Kielce, Poland; 2Holy Cross Cancer Center, Tumors Markers Department, PL-25734 Kielce, Poland

**Keywords:** methylisothiazolinone, boron-doped diamond electrode, voltammetry

## Abstract

The electrochemical properties of methylisothiazolinone (MIT), the most widely used preservative, were investigated by cyclic (CV) and differential pulse voltammetry (DPV) to develop a new method for its determination. To our knowledge, this is the first demonstration of a voltammetric procedure for the determination of MIT on a boron-doped diamond electrode (BDDE) in a citrate–phosphate buffer (C-PB) environment. The anodic oxidation process of methylisothiazolinone, which is the basis of this method, proved to be diffusion-controlled and proceeded with an irreversible two-electron exchange. The radical cations, as unstable primary products, were converted in subsequent chemical reactions to sulfoxides and sulfones, and finally to more stable final products. Performed determinations were based on the DPV technique. A linear calibration curve was obtained in the concentration range from 0.7 to 18.7 mg L^−1^, with a correlation coefficient of 0.9999. The proposed procedure was accurate and precise, allowing the detection of MIT at a concentration level of 0.24 mg L^−1^. It successfully demonstrated its suitability for the determination of methylisothiazolinone in household products without the need for any separation steps. The proposed method can serve as an alternative to the prevailing chromatographic determinations of MIT in real samples.

## 1. Introduction

Many groups of water-based products, including cosmetics, personal care products, household products, and pharmaceuticals, require protection from microorganisms (fungi and bacteria) to ensure their properties, suitability, and safety for users and to extend their shelf life. Such functions are performed by preservatives. These are the compounds that, when added at relatively low concentrations to protected objects, block, destroy, inactivate and prevent the action of harmful organisms by chemical and/or biological means. Preservatives are characterized by their diverse chemical structure. Literature data show that the most numerous group of preservatives comprises isothiazolinones and other nitrogen compounds. Isothiazolinone derivatives, such as methylisothiazolinone (2-methyl-4-isothiazolin-3-one, MIT), methylchloroisothiazolinone (5-chloro-2-methyl-4-isothiazolin-3-one, CMIT), and 4,5-dichloro-2-octyl-4-isothiazolin-3-one (DCOIT) (Figure 1), are used in various types of consumer products such as cosmetics (moisturizers, eye shadows, and make-up removers), hair and skin care products [1,2,3], adhesives [4], water-based paints [5], hydraulic fracturing fluids [6], biodiesels [7], reverse osmosis systems for water desalination [8], cooling water treatment [9], household products [3], and in the textile and paper industries [10]. This widescale use is due to their excellent biocidal properties at low concentrations, which in turn, are related to the active and oxidation-prone sulfur molecule.

In the past, a mixture of methylisothiazolinone and methylchloroisothiazolinone in a ratio of 1:3 (trade names: Kathon CG, Euxyl 400) was mainly used, but it was not until MIT started to be used in higher concentrations due to its allergenic effect that it was supposed to be a better-tolerated chemical compound [11,12,13]. However, it has also been shown in the literature to be highly allergenic [14,15]. Occupational cases of allergy to this compound involve medical personnel, painters, turners, mechanics, catering workers, cleaning staff, hairdressers and beauticians [16,17]. In 2013, MIT was announced as allergen of the year by the American Contact Dermatitis Society [18]. For this reason, the use of MIT in leave-on products was banned in 2016 [19]. In turn, in 2017, the European Commission published a new regulation limiting the use of MIT in rinse-off products to a maximum concentration of 0.0015% [20].

The widespread use of isothiazolinones, as well as numerous reports in the literature on their harmful effects on the human body, make it important to quantify them in various matrices. The determination of isothiazolinones is most commonly performed by high-performance liquid chromatography (HPLC) [2,3,5,21,22,23,24,25,26,27,28] and sometimes by ultra-high-performance liquid chromatography (UHPLC) [1,10,29,30]. Gas chromatography (GC) is a much less commonly used technique for this purpose [28]. Mass spectrometry (MS) [25,28] and tandem mass spectrometry (MS/MS) [1,3,5,10,24,26,27,31] predominate as detection methods in the chromatographic analysis of these biocides. Spectrophotometric detectors in the UV range are also frequently used [2,21,22,23,29,30,32]. To our knowledge, only one method for the determination of MIT by HPLC coupled with electrochemical detection (ECD) has been developed to date. Abad-Gil et al. [33] used their previous studies on the voltammetric determination of MIT with a the gold electrode [34] to develop a procedure for the simultaneous chromatographic determination of antimicrobial agents in cosmetics. The *LOD* value of 30 µg L^−1^ obtained for MIT is lower than those previously reported for HPLC with UV detectors [2,21,29,30] and comparable for MS/MS [1]. The advantage of chromatographic techniques is excellent selectivity and sensitivity, allowing the simultaneous determination of many analytes and achieving low detection limits. Their serious drawbacks are expensive and complex equipment and the consumption of many reagents necessary to prepare the sample for analysis (clean-up, extraction or/and derivatization) and elution. Analytical methods based on voltammetry may be an attractive alternative for the determination of isothiazolinones. They do not require expensive equipment, and sample preparation is often limited to dissolving a sample in a suitable medium. They are equal in sensitivity and precision to chromatographic techniques [34,35].

To our knowledge, only four papers published to date deal with the voltammetric determination of isothiazolinones [34,35,36,37]. According to Abad-Gil et al. [34], MIT can be successfully determined on a gold anode in phosphate buffer solutions at pH 6 using square wave voltammetry (SWV) with a linear response up to 53 mg L^−1^. Using adsorptive stripping voltammetry (SWAdSV), it was possible to analyze samples containing MIT at very low concentrations ranging from 0.027 to 0.12 mg L^−1^. The *LOD* and *LOQ* values obtained were 2.8 and 9.4 mg L^−1^ for SWV and reached significantly lower values of up 0.008 and 0.027 mg L^−1^ for SWAdSV. The same group of researchers developed a voltammetric method for the determination of MIT in cosmetic and water samples using a screen-printed electrode (SPCE) modified with poly(diallyldimethylammonium) (PDDA) nanocomposite membranes containing a gold nanoparticles (AuNp) [35]. The experiments were performed using the cyclic voltammetry (CV) technique in 0.1 M NaOH solutions. This method allowed the determination of MIT in a concentration range from 8.7 to 36 mg L^−1^ and with the *LOD* and *LOQ* values of 2.6 and 8.7 mg L^−1^, respectively. No interferences were observed in the presence of many cations and organic compounds.

A simple differential pulse voltammetric (DPV) method for the quantification of MIT and CMIT in cosmetics using the standard additional method developed by Wang et al. [36]. The determinations were preceded by the extraction of the analytes with dichloromethane. It was shown that the best results were obtained using carbon fiber (CF) microelectrode in solutions of 0.1 M LiClO_4_ (pH 6.04). The linearity of the method was found over the range of 2–260 mg L^−1^ and 4–230 mg L^−1^ for MIT and CMIT, respectively.

An interesting indirect DPV method based on the interaction of cysteine with MIT and CMIT for their voltammetric determination on a glassy carbon electrode (GCE) in phosphate buffer solutions was proposed by Montoya et al. [37]. The specificity of these interactions and low peak potential values minimized the interference from matrix components. The determinations were based on the decrease in the cysteine oxidation signal with increasing biocide concentration.

The limited number of procedures developed for the voltammetric determination of isothiazolinones is probably due to little knowledge of their electrochemical properties in different environments. As far as we know, no systematic studies of the electrochemical properties of isothiazolinones have been conducted to find the optimal conditions for their voltammetric determination in real samples. The anodic oxidation of these biocides, most commonly MIT, CMIT, and DCOIT, has mainly been studied during the development of methods for their electrochemical degradation [38,39,40,41] or determination [34,37]. The working electrodes made of various materials such as Ti/SnO_2_-Sb/PbO_2_ [38], Sb_2_O_3_/α, β-PbO_2_ [39], boron-doped diamond (BDD) [40], and carbon fiber felt (CFF) [41] have been used. All the oxidative degradation pathways described involved a ring-opening reaction and the breaking of the weak sulfur–nitrogen bond in MIT molecules is postulated [38,40], but cleavage of the carbon–carbon double bond is also considered [39,41]. Subsequent oxidation, hydrolysis, and loss of sulfur, as well as its conversion to SO_4_^2−^ lead to the formation of many compounds, including N-containing products with fewer carbon atoms, organic acids (acetic and formic), which can eventually be slowly converted into CO_2_, NO_3_^−^, H_2_O, and additionally into HCl during CMIT oxidation [38,39,40]. In the case of a CFF-based flow-through electrode system (FES) the electrochemical degradation of MIT was via direct anodic oxidation, where the organic sulfur was oxidized to an unstable sulfoxide or sulfone structure [41]. The possibility of direct oxidation of isothiazolinones on the surface of a BDD anode was demonstrated by Kandavelu et al. [40]. The CV curves recorded in the presence of these compounds consisted of a well-defined anodic peak at 1.72 V and a shoulder at 1.58 V vs. SCE. The shape of the signals resulted from the overlapping of the two characteristic oxidation peaks of CMIT and MIT, respectively. The absence of any cathodic peaks indicated irreversible oxidation of the biocides. The same voltammetric technique was used to study the electrochemical properties of MIT on a gold electrode in phosphate buffer solutions in the pH range of 2–10 by Abad-Gil et al. [34]. The anodic oxidation of this compound was shown to be an irreversible and diffusion-controlled process in which two electrons and two protons were exchanged to yield sulfoxides and, ultimately, sulfones. Unlike other cited researchers [38,39,40,41], the authors of [34] did not include the ring-opening reaction in the proposed MIT oxidation mechanism. The results obtained were applied to the voltammetric determination of MIT [34] and to the construction of an electrochemical detector in HPLC for the detection of various antimicrobial agents, including MIT in cosmetic products [33]. The two-electron anodic oxidation processes of MIT, CMIT, and DCOIT in phosphoric acid and phosphate buffer solutions (pH 2–11) on GCE with their ring-opening were postulated by Montoya et al. [37]. The CV curves recorded in the studied solutions showed a poorly shaped MIT oxidation peak at a potential of about 1.5 V vs. Ag/AgCl. When CMIT was anodically oxidized, two overlapping peaks appeared at the same potential as the MIT peak. The oxidation of DCOIT in this environment occurs at potentials higher than those characteristic for MIT and CMIT (about 1.7 V vs. Ag/AgCl). Considerably better-shaped peaks were obtained when the DPV technique was used. The anodic peak potentials of all three biocides remain constant up to pH 6 and decrease slightly above this pH value. This indicates that protons are not involved in the electrode reaction. The results show that all the biocides studied undergo a two-electron anodic ring-opening oxidation. The oxidation process is of *ECE* type in which the rate-determining step is a chemical reaction placed after the first partially reversible electron transfer.

To our knowledge, there has been no work to date on the use of a boron-doped diamond electrode (BDDE) for the voltammetric determination of isothiazolinones. No have citrate-phosphate buffer solutions been used for this purpose. Therefore, this study aims to investigate the electrochemical properties of MIT—the major isothiazolinone on BDDE in aqueous buffer solutions—to test the possibility of developing a voltammetric method for its determination in real samples.

## 2. Results and Discussion

### 2.1. Selection of the Best Voltammetric Conditions

In order to obtain an optimal environment for the investigation of the electrochemical properties of MIT, as well as for its voltammetric determination, the DPV curves were recorded in various types of buffer solutions in buffer-specific pH ranges (Figure 1). Britton-Robinson (B-RB), acetate (AcB), phosphate (PB), citrate (CB), and citrate–phosphate (McIlvaine, C-PB) aqueous buffers were tested. The position and shape of the curves, as well as the peak currents, were found to be closely dependent on the type of buffer and its pH. In each of the buffer solutions tested, it was found that there was a pH at which the peak current, *I*_p_ reached a maximum value. The curves corresponding to such conditions are shown in Figure 1B.

The best-shaped curves with the highest current intensity to guarantee high sensitivity of future determinations were observed in the McIlvaine buffer. In all other buffer solutions, asymmetric curves (CB) or much lower peak currents, *I*_p_ (B-RB, AcB, PB) were obtained. The data presented in Figure 1 indicate that optimal conditions can be obtained in C-PB solutions at pH 5.6. The oxidation of the analyte in this environment proceeds in one step giving a well-defined peak at a potential of about 1.535 V vs. Ag/AgCl. The DPV peak width at half height, *W*_1/2_ of 0.130 V is well above the theoretical predicted for reversible exchange of electron, *n* (*W*_1/2_ = 0.0904/*n* V at 25 °C [42]). This indicates an irreversible anodic oxidation of MIT. The same was observed in other buffers. Therefore, the McIlvaine buffer, pH 5.6, was chosen for further studies. It should be noted that the concentration of CP-B buffer components has no significant effect on the magnitude and shape of the MIT signal.

The next part of the experiments was the selection of the optimal working electrode material. For this purpose, DPV curves were recorded in the chosen composition of the solution on electrodes made of gold (Au), platinum (Pt), glassy carbon, and boron-doped diamond (Figure 2). When the electrode material was gold, to prevent its oxidation, the curves were recorded at 1.4 V. The only signal appeared at a potential of about 1 V. However, it cannot be associated with MIT because it also occurs in the case of the DPV curve recorded for the supporting electrolyte itself. This signal may be related to the formation of an oxide layer on the surface of the gold electrode. In the presence of MIT, only a slight increase in this signal was observed, which may suggest that oxidation of methylisothiazolinone takes place on the oxide layer. Another reason for the lack of a clear signal for MIT oxidation may be the blocking of the electrode surface by the analyte caused by the affinity of sulfur to gold. The signals recorded on GCE and platinum at potentials of about 1.7 and 1.8 V, respectively, are poorly shaped, unreproducible, and thus unsuitable for analytical purposes. The best-shaped and symmetrical MIT oxidation curve with the lowest residual currents was obtained on a boron-doped diamond electrode (Figure 2). In addition, this material provides the best repeatability of successively recorded curves. Therefore, BDDE with a diameter of 3 mm was used in further studies.

In order to optimize the measurement conditions, the influence of the operating parameters of the DPV technique was also studied. They should ensure the best-shaped curves obtained, high resolution of signals, and the maximum peak current guaranteeing the highest sensitivity and, thus, achieving low detection limits. The operating parameters of the DPV technique and their test values are shown in Table 1. It was noted that *I*_p_ increased gradually as the DPV amplitude (d*E*) increased. Since values of d*E* greater than 50 mV cause a sharp increase in *W*_1/2_, and thus reduce the resolution of the signals, an amplitude of 50 mV was chosen for further study. Considering the shape and MIT oxidation peak current, the potential step of 5 mV and the pulse width of 80 ms were selected.

### 2.2. Electrochemical Properties of MIT in C-PB

The electrochemical properties of MIT on BDDE were performed in an experimentally selected C-PB buffer (pH 5.6). For this purpose, cyclic voltammetry was used. This voltammetric technique is usually applied to investigate the reduction and oxidation processes of both organic and inorganic species and to study their mechanisms. The analysis of CV curves allows obtaining information on the characteristics of electrode processes and accompanying homogeneous reactions, electrochemical properties of the analyzed substances, as well as their reduction or oxidation products.

The preliminary investigations indicate that MIT undergoes anodic oxidation in the applied environment giving a well-shaped, single peak at a potential above 1.5 V vs. Ag/AgCl. The selected CV curves of MIT recorded in the potential window from −1.2 to 2.0 V are shown in Figure 3. The position of the signal is similar to those reported in the literature and observed in other buffer solutions [36,37,40]. A characteristic feature of the CV curves recorded on BDDE, both in the absence and presence of MIT, were very small residual currents. The observed signal position corresponds to the DPV curves recorded in the same buffer solution (*E*_p_ = 1.535 ± 0.005 V vs. Ag/AgCl; Figure 1). The unlimited current increase above 1.8 V is related to the oxidation of the buffer components. As Figure 3 shows, no corresponding reduction peak is observed over a wide cathodic potential range. The cathodic signal did not appear even when the scan rate increased, and the polarization direction, *E*_λ_ was reversed near the anodic peak (Figure 3, inset). The reason for this can be due to a completely irreversible electrode reaction or instability of the primary MIT oxidation product, which may participate in the following homogeneous reaction near the electrode surface. This chemical reaction can transform it into non-electroactive products or those that were not reducible in the potential range available in the environment studied. Other researchers have also observed the absence of a cathodic peak on the CV curves recorded in the presence of these compounds [37,40].

In the next stage of the experiments, the influence of the scan rate, *v* in the range of 0.0062 to 0.5 V s^−1^, on the MIT peak current and its peak potential was investigated. The CV curves shown in Figure 4 indicate the peak currents increase as the potential scan rate increases. According to the Randles-Sevcik equation, a linear relationship was observed between the anodic peak current and the square root of the scan rate (*I*_p_ (μA) = 38.26 *v*^1/2^ (V s^−1^)^1/2^ + 3.34, *r* = 0.9999, Figure 4, inset A). This indicates that the electrode process is controlled by the diffusion of the analyte onto the surface of the working electrode. This conclusion is confirmed by the plot of log *I*_p_ vs. log *v* (log *I*_p_ (μA) = 0.41 log *v* (V s^−1^) + 1.60, *r* = 0.9995, Figure 4, inset B), whose slope of 0.41 is close to the theoretical value of 0.5 characterizing diffusion-controlled processes [42,43].

When recording successive scans without cleaning the electrode surface, a small and gradual decrease in peak currents was observed. The largest changes were noticed between the first and second cycles (a 15% decrease in the peak current). Subsequently, the peak currents decreased by 8% and 2% between cycles 2 and 3, and 3 and 4, respectively. This indicates that the oxidation of MIT was accompanied by passivation of the electrode and, thus, a reduction in its active surface area. The peaks in successive cycles reached a stable height. Since the anodic oxidation of MIT is diffusion-controlled, the decrease in the peak current can be attributed to the partial adsorption of the final products of the electrode process. The reproducibility of the successively recorded curves was restored by applying cathodic polarization in the region of the hydrogen evolution potential (−1.2 V by 30 s). A similar decrease in the anodic peak current, corresponding to the oxidation of MIT and CMIT on BDDE in aqueous Na_2_SO_4_ solutions, was observed by Kandavelu et al. [40]. However, the electrode activity was restored in the region of water decomposition potential (>2.3 V vs. SCE).

The CV curves shown in Figure 4 indicate that the peak potential of the MIT oxidation, *E*_p_, slightly shifted toward more positive with an increase in the potential scan rate. This indicates that the anodic oxidation of MIT is not reversible. To check the reversibility of the electrode process, a criterion based on the difference between the peak potential and the potential corresponding to 1/2 the peak current (*E*_p_ − *E*_p/2_) was applied. The theoretical value characteristic of the reversible process is 0.0564/*n* V [42] at 298 K. The experimental values are higher than the theoretical ones, and they increased with an increasing potential scan rate from 0.078 V to 0.093 V (for *v* = 0.025 and 0.500 V s^−1^, respectively). This confirms that the anodic oxidation of MIT is irreversible. In this case, the relationship between *E*_p_ and *E*_p/2_ is described by the equation *E*_p_ − *E*_p/2_ = 0.0477/*n*_e_*α* V, where α is the transfer coefficient, and *n*_e_—the number of electrons exchanged in a rate-determining step [42,43]. Since most often *n*_e_ = 1, the value of α can be easily obtained from this equation [43]. An *α* value in the range of 0.61 to 0.51 was obtained for scan rates of 0.025 to 0.500 V s^−1^. The irreversibility of this electrode reaction is confirmed by the relationship between the peak potential and the log of the scan rate shown in Figure 4, inset C. In this case, the peak potential should be a linear function of *v* and described by the equation: *E*_p_ (V) = (2.303*RT*/2*αnF*) log[*v* (V s^−1^)] + const. [42], where *n* is the total number of electrons transferred. The theoretical value of the slope of this relationship at 298 K should be 0.030/*αn* V per decade change in scan rate. The experimentally obtained dependence (Figure 4, inset C) is described by the equation: *E*_p_ (V) = 0.03093 log *v* + 1.624 (*r* = 0.9997). The slope of this curve is close to the theoretical one predicted for a 2-electron irreversible electrode reaction at the experimental value of the transfer coefficient, α close to 0.5. The anodic oxidation of MIT thus proceeds with an irreversible exchange of two electrons.

The contribution of protons in the electrode process was checked for the effect of pH on the peak potentials of the CV curves in the range from 2.4 to 6.0. The obtained results indicate that an increase in pH caused only a slight shift in the oxidation peak potentials toward less positive values (data not shown). The obtained plot of peak potential, *E*_p_ vs. pH is linear and is expressed by the equation: *E*_p_ (V) = 1.605 − 0.0041 pH (*r* = 0.998). A slight slope of this relationship (0.0041 V/pH) deviates significantly from the theoretical one expected for an equal number of protons and electrons involved in the electrode reaction (0.0591 V/pH). These results indicate that the anodic oxidation of MIT in the tested pH range proceeds without proton exchange. They are consistent with the results obtained by Montoya et al. [37] on GCE in phosphate buffer solution.

### 2.3. Mechanism

The results obtained indicate that the anodic oxidation of MIT on BDDE in a citrate-phosphate buffer solution is diffusion-controlled and proceeds with an irreversible exchange of two electrons. Protons, however, do not participate in this heterogeneous reaction. In addition, the primary product is unstable and undergoes an irreversible chemical reaction near the surface of the working electrode. This is evidenced by the absence of cathodic peaks in the CV curves (Figure 3 and Figure 4). Taking into account the results obtained, a probable mechanism for the anodic oxidation of MIT in C-PB was proposed, which is shown in Figure 2. Given the susceptibility of sulfur to oxidation and its easy access to the electrode surface, the electrode process can proceed with its participation. The irreversible loss of the first electron (*E*_i1_) leads to the formation of an unstable radical cation. This primary product can react with a water molecule, leading to the formation of sulfoxide. The sulfoxide thus formed can undergo a second irreversible stage of one-electron anodic oxidation (*E*_i2_) to a radical cation, which is then transformed into a sulfone molecule under the influence of water. The oxidation of MIT to sulfoxides and sulfones has been suggested in studies of their direct electrochemical degradation on carbon fiber felt [41] and on a gold electrode [33,34]. Both sulfoxides and sulfones are unstable [41] and can undergo subsequent chemical reactions, including breaking the weakest sulfur–nitrogen bond and opening the ring [38,40]. These homogeneous chemical reactions can lead to the loss of a sulfur atom and its conversion to SO_4_^2−^ ions and the formation of many stable compounds, such as N-containing products with fewer carbon atoms, NO_3_^−^ ions, and organic acids (acetic and formic) [38,39,40,41]. It is worth noting that the presented mechanism is only probable and consistent with the literature data. Our laboratory has not been able to identify the primary and final products.

### 2.4. Validation of the MIT Determination Method

The applicability of the proposed voltammetric method for the determination of methylisothiazolinone was examined by measuring the peak current as a function of analyte concentration using the optimized parameters of the DPV technique. Figure 5 and inset A display the DPV curves at different MIT concentrations. It was found that the peak current increases linearly with increasing its concentration in the range of 0.7–18.7 mg L^−1^ (Figure 5B). The calibration curve was described by the equation: *I*_p_ (μA) = (1.147841 ± 0.032) × *c* (mg L^−1^) − (0.095063 ± 0.087), *r* = 0.9999 (n = 25). Based on the standard deviation of the intercept *S*_b_, and the slope *a* of the calibration curve, the detection limit (*LOD* = 3.29 *S*_b_/*a*) of MIT was determined to be 0.24 mg L^−1^. This is well below the legal level allowed in cosmetics [20]. The *LOD* value is lower than those obtained by HPLC-DAD [2,21], square wave [34], cyclic voltammetry [35], comparable to the value for UHPLC-PDA [30] and higher for GC-MS [28], HPLC-MS/MS [27] and adsorptive stripping voltammetry [34]. Additionally, McIlvaine buffer pH 5.6 provides one of the wider useful concentration ranges compared to the literature data [21,29,34,35].

The intra-day reproducibility was determined by recording DPV curves (n = 10) in 0.75 mg L^−1^ MIT solution. The obtained relative standard deviation (*RSD*) of the peak current of 0.6% demonstrates the excellent repeatability of the developed method. The inter-day precision of the results was determined by measuring *I*_p_ over a period of 5 consecutive days using the same solution, and the *RSD* value obtained was 1.1%.

To evaluate the selectivity of the procedure, the influence of possible interfering species, such eugenol, methyl paraben, 4-hydroxybenzoic acid, and also inorganic ions K^+^, Na^+^, Mg^2+^, Ca^2+^, SO_4_^2−^, NO_3_^−^ which can be found in cosmetics and household products along with MIT, were investigated. The impact of the 10-fold excess of interferences concentration on the peak currents of MIT did not exceed 5%. These results indicate that it is possible to determine methylisothiazolinone in the presence of these interferents.

The developed method was verified by control determinations (the procedure described in Section 3.3). The methylisothiazolinone content of a selected household product was determined in the same way. The example curves are shown in Figure 6A. The content of the analyte in the solution was determined using mEAlab 2.1 software based on the dependence of the peak current on the concentration of the standard (Figure 6B). The resulting concentration in the real sample was then converted to the content in 1 g of product and statistically analyzed (Table 2). 

Based on the analysis of the results of control determinations and those obtained in the solution of the selected household product, it was concluded that the developed procedure for the voltammetric determination of methylisothiazolinone was accurate (recovery, *R* = 99.0–99.5%, without Perwoll matrix) and precise (*RSD* ≤ 1.0%). The MIT content in control solutions with the matrix is consistent with the sum of the amounts introduced and determined in the matrix.

## 3. Materials and Methods

### 3.1. Reagents

All the reagents were of high quality and were used as received: 2-methyl-1,2-thiazol-3(2H)-one (MIT, ≥95%, Sigma-Aldrich, St. Louis, MO, USA), sodium acetate (CH_3_COONa, AcNa, anhydrous, >99.0%, Merck, Darmstadt, Germany), sodium citrate (Na_3_C_6_H_5_O_7_ × 2H_2_O, ≥99.0%, Sigma-Aldrich, St. Louis, MO, USA), disodium hydrogen phosphate (Na_2_HPO_4_, anhydrous, ACS, Reag. Ph Eur, Merck, Darmstadt, Germany), potassium dihydrogen phosphate (KH_2_PO_4_, anhydrous, ACS, Reag. Ph Eur, Merck, Darmstadt, Germany), sodium hydroxide (NaOH, p.a., 98.8%, Pol-Aura, Olsztyn, Poland), glacial acetic acid (CH_3_COOH, AcH, p.a. ACS, Merck, Darmstadt, Germany), phosphoric acid (H_3_PO_4_, 85 wt.%, Sigma-Aldrich, St. Louis, MO, USA), citric acid (H_3_C_6_H_5_O_7_, ≥99.5%, Sigma-Aldrich, St. Louis, MO, USA), sulfuric (VI) acid (H_2_SO_4_, p.a., 95 wt.%, Chempur, Piekary Slaskie, Poland). Perwoll washing liquid (Henkel, Wien, Austria) was used as an example of the MIT determination in the real sample. 

Britton–Robinson buffer (B-RB) solutions were prepared by mixing phosphoric acid, boric acid, and acetic acid (all at a concentration of 0.04 mol L^−1^). Their pH ranged from 2.0 to 10.0 and was adjusted with 0.2 mol L^−1^ NaOH. A series of McIlvaine buffers (citrate-phosphate buffers, C-PB) with pH in the range of 2.4–8.3 was obtained by mixing different volumes of 0.4 mol L^−1^ disodium hydrogen phosphate and 0.2 mol L^−1^ citric acid. Citrate buffers (CB) were prepared by mixing 0.2 mol L^−1^ solutions of citric acid and sodium citrate in different volumes. The pH range of these buffer solutions was from about 3.0 to 5.9. Phosphate buffers (PB) were prepared by mixing 0.2 mol L^−1^ solutions of disodium hydrogen phosphate and potassium dihydrogen phosphate in various amounts. The pH range of these buffer solutions was from about 5.9 to 7.5. Acetate buffers (AcB) were obtained by mixing 0.2 mol L^−1^ solutions of acetic acid and sodium acetate to obtain the desired pH in the range of 3.8 to 5.7. Double-distilled water was used to prepare the buffer solutions.

### 3.2. Apparatus

The voltammetric experiments were performed using a computer-controlled Model M161 electrochemical analyzer cooperating with EALab 2.1. software (mtm-anko, Cracow, Poland). A three-electrode glass measuring cell with a volume of 10 mL consisted of a boron-doped diamond electrode, BDDE with a diameter of 3 mm (BioLogic Science Instruments, Seyssinet-Pariset, France), a platinum wire (BASi, West Lafayette, IN, USA), and Ag/AgCl with 3 mol L^−1^ KCl (Mineral, Warsaw, Poland) as the working, counter and reference electrode, respectively. Some experiments were carried out using working electrodes of platinum, gold, and glassy carbon with a diameter of 3 mm (each BASi, West Lafayette, IN, USA).

The pH of the buffer solutions was measured using a CX-732 multifunction meter with a sensor consisting of a glass indicator electrode and an Ag/AgCl reference electrode (Elmetron, Zabrze, Poland).

All experiments were conducted at a constant temperature of 25 ± 1 °C.

### 3.3. Electrochemical Measurements

The EC behavior of MIT on BDDE was studied by cyclic and differential pulse voltammetry. Prior to all the voltammetric experiments, the BDDE surface was mechanically polished with 0.01 μm alumina powder slurry on a polishing cloth, then sonicated in distilled water and dried. In the next stage, the electrode surface was cathodically activated in 1 mol L^−1^ H_2_SO_4_, at a potential of −2.4 V, for a period of 5 min [44,45]; this procedure was repeated each day before starting all voltammetric measurements. The BDDE surface did not require additional treatment between the registrations of the voltammetric curves. This pre-treatment method guaranteed the highest activity of the electrode surface and, thus, the highest MIT oxidation currents, and the best reproducibility of voltammetric curves. The CV curves were recorded at scan rates from 6.2 to 500 mV s^−1^ in the potential range of −1.2 to 2.0 V vs. Ag/AgCl. To improve the electrode response, the DPV technique parameters, such as a potential step, the pulse width, and the pulse amplitude, were optimized. Their values are shown in Table 1. The DPV curves were recorded in the potential range of 1.0 V (beginning) to 1.8 V (end) vs. Ag/AgCl. The reproducibility of the successively recorded curves was restored by applying cathodic polarization in the area of the hydrogen evolution potential (−1.2 V by 30 s).

The main voltammetric measurements were carried out on BDDE in an experimentally selected citrate-phosphate buffer solution at pH 5.6. The calibration procedure was based on the DPV curves of MIT oxidation in the concentration range of 0.24 to 27.1 mg L^−1^. Test solutions with the desired amount of MIT were obtained by dissolution in an appropriately defined pH buffer.

The quantitative analysis was carried out by the multiple standard addition method. Optimal determination conditions were obtained using the MIT standard at a concentration that guaranteed a significant increase in the analytical signal. The appropriate standard was added in portions of 50 μL to 2.0 mL to the solution of the test preparations. The sample preparation procedure was limited only to dissolving the product in a solution of citrate–phosphate buffer at pH 5.6. The test product was introduced in such an amount as to obtain an analyte signal suitable for quantitative analysis.

The reliability of the procedure was checked by control determinations at two MIT concentration levels of 0.96 and 4.98 mg L^−1^, without and with the addition of a matrix (about 1 g of the Perwoll product containing MIT). The DPV curves were recorded in the stock solution and after the successive additions of the methylisothiazolinone standard. The constructed relationship *I*_p_ = f(*c*_MIT_) was used to determine the concentration of the analyte. The procedure was repeated five times, and the results were subject to statistical analysis.

## 4. Conclusions

The obtained results indicate that citrate-phosphate buffer solution at pH 5.6 is a suitable medium for studying the electrochemical properties of methylisothiazolinone and its determination. The process of the anodic oxidation of MIT is characterized as diffusion-controlled and proceeds with an irreversible exchange of two electrons. Based on this electrode reaction, a new voltammetric method was developed using DPV on a boron-doped diamond electrode for the determination of MIT. To the best of our knowledge, this is the first procedure using C-PB and BDDE for this purpose. A linear relationship was obtained between the MIT concentration and the current response in the range of 0.7 to 18.7 mg L^−1^, with a detection limit of 0.24 mg L^−1^, which is lower than those obtained by square wave and cyclic voltammetry presented in the literature. The developed procedure is accurate and precise, and its utility was successfully demonstrated in the determination of methylisothiazolinone in household products. Its advantage is the simplicity of sample preparation, which does not require any separation steps, thus reducing the time of analysis. In addition, the use of aqueous buffer solutions and small sample volumes make the proposed method compliant with the principles of green chemistry. Consequently, the developed procedure can be a useful tool for quality control analysis of products containing MIT.

## Data Availability

The data presented in this study are available on request from the corresponding author.

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
