# Peer review of "Electrochemical Characterization and Voltammetric Determination of Methylisothiazolinone on a Boron-Doped Diamond Electrode"

_molecules, 2022, doi:10.3390/molecules27249013_

Round 1

Reviewer 1 Report

The authors have reported the Electrochemical Characterization and Voltammetric Determination of Methylisothiazolinone on a Boron-Doped Diamond Electrode. However, if issues surround the manuscript like

1. Abstract: Line 16-21, if a community has no assess to cyclic (CV) , what other means can  MIT be detected at same low concentrations?

2. Introduction: it is too lengthy, the author(s) should revised to reduce 

3. Line 211-214: What could be responsible for the best-shaped curves in the use of McIlvaine buffer than other types of buffers?

4. Line 221-225: What guided the choice of electrode for this experiments.? 

5. Scheme 2: what is the expected final product. ?

Author Response

We would like to thank the Reviewer for their thoughtful review of the manuscript. It raise important issues and its inputs are very helpful for improving the manuscript. We agree with all comments and we have revised our manuscript accordingly.

We respond below in detail to each of the comments. In addition, we include how we have revised things. We hope that the reviewer will find our responses to their comments satisfactory.

Please, find below your comments repeated in italics and our responses inserted after each comment.

Yours sincerely,

Magdalena Jakubczyk

Response to comments from Reviewer

  1. Abstract: Line 16-21, if a community has no assess to cyclic (CV), what other means can MIT be detected at same low concentrations?

The abstract has been supplemented with sentence:

“Performed determinations were based on DPV technique.”

  1. Introduction: it is too lengthy, the author(s) should revised to reduce

We agree with the reviewer that introduction is too lengthy. The introduction has been shortened accordingly. Information necessary to interpret own results has been left.

  1. Line 211-214: What could be responsible for the best-shaped curves in the use of McIlvaine buffer than other types of buffers?

This problem is difficult to explain. Well-shaped curves and high currents are probably related to the specific interactions of the analyte with the buffer components (citrate and/or phosphate ions). This problem is irrelevant from the point of view of quantitative research.

  1. Line 221-225: What guided the choice of electrode for this experiments?

Explanation was incorporated in the text (pg. 5-6, lines 220-237):

The next part of the experiments was the selection of the optimal working electrode material. For this purpose, DPV curves were recorded in the chosen composition of the solution on electrodes made of gold (Au), platinum (Pt), glassy carbon and boron-doped diamond (Figure 2). When the electrode material was gold, to prevent its oxidation, the curves were recorded at 1.4 V. The only signal appeared at a potential of about 1 V. However, it cannot be associated with MIT, because it also occurs in the case of the DPV curve recorded for the supporting electrolyte itself. This signal may be related to the formation of an oxide layer on the surface of the gold electrode. In the presence of MIT, only a slight increase in this signal was observed, which may suggest that oxidation of methylisothiazolinone takes place on the oxide layer. Another reason for the lack of a clear signal for MIT oxidation may be the blocking of the electrode surface by the analyte due to the affinity of sulfur for gold. On GCE and platinum, the signals observed at potentials of about 1.7 and 1.8 V, respectively, are distorted, unreproducible and unsuitable for analytical purposes. As can be seen, the best-shaped and symmetrical MIT oxidation curve with the lowest residual currents was obtained on a boron-doped diamond electrode. In addition, this material provides the best repeatability of successively recorded curves. Therefore, BDDE with a diameter of 3 mm was used in further studies.

  1. Scheme 2: what is the expected final product?

Explanation was incorporated in the text (pg. 10, lines 362-371):

The oxidation of MIT to sulfoxides and sulfones has been suggested in studies of their direct electrochemical degradation on carbon fiber felt [41] and on a gold electrode [33,34]. Both sulfoxides and sulfones are unstable [41] and can undergo subsequent chemical reactions, including breaking the weakest sulfur-nitrogen bond and opening the ring [38,40]. These homogeneous chemical reactions can lead to the loss of a sulfur atom and its conversion to SO42- ions and the formation of many stable compounds, such as N‑containing products with fewer carbon atoms, NO3- ions, and organic acids (acetic and formic) [38–41]. It is worth noting that this is only plausible and consistent with the literature data for the mechanism of this process. Our laboratory has not been able to identify its primary and final products.

Reviewer 2 Report

In this work electrochemical properties of methylisothiazolinone have been studied by using a boron-doped diamond electrode.  Cyclic and differential pulse voltammetry methods applied and results have been discussed. When evaluated following comments should be considered:

1.      English language of the manuscript should be more scientific and should be revised in a scientific way. For example phrases like “to evaluate the selectivity of our procedure” or “as can be seen from the” should be avoided.

2.      Introduction part seems too long and more detail is given which is not necessary for the work carried. Introduction should be summarized in a meaningful way. For example in page 2 line 64 unnecessary details are given related with gradient elution solvents and similar other examples exist.

3.      Choramotographic drawbacks given in introduction should also be summarized in one sentence in a much more meaningful way.

4.      Page 3 line 91 referance should be included for related chromatographic techniques.

5.      Page 10 line 386 the phrase “widest ranges of linearity” should be defined and explained scientifically.

6.      Conclusion should be improved by means of novelty of the electrodes used and the method.

Author Response

We would like to thank the Reviewer for their thoughtful review of the manuscript. It raise important issues and its inputs are very helpful for improving the manuscript. We agree with all comments and we have revised our manuscript accordingly.

We respond below in detail to each of the comments. In addition, we include how we have revised things. We hope that the reviewer will find our responses to their comments satisfactory.

Please, find below your comments repeated in italics and our responses inserted after each comment.

Yours sincerely,

Magdalena Jakubczyk

Response to comments from Reviewer

  1. English language of the manuscript should be more scientific and should be revised in a scientific way. For example phrases like “to evaluate the selectivity of our procedure” or “as can be seen from the” should be avoided.

We agree with the reviewer that some wording was not correct. We corrected the English language of the manuscript as suggested by the reviewer. The text has been verified by a certified speaker.

  1. Introduction part seems too long and more detail is given which is not necessary for the work carried. Introduction should be summarized in a meaningful way. For example in page 2 line 64 unnecessary details are given related with gradient elution solvents and similar other examples exist.

We agree with the reviewer that introduction is too lengthy. The introduction has been shortened accordingly. Information necessary to interpret own results has been left.

  1. Chromatographic drawbacks given in introduction should also be summarized in one sentence in a much more meaningful way.

The reviewer's comment has been taken into account.

  1. Page 3 line 91 reference should be included for related chromatographic techniques.

The reviewer's comment has been taken into account.

  1. Page 10 line 386 the phrase “widest ranges of linearity” should be defined and explained scientifically.

The sentence has been supplemented with literature data:

“Additionally, McIlvaine buffer pH 5.6 provides one of wider useful concentration ranges compared to literature data [21,29,34,35].”

  1. Conclusion should be improved by means of novelty of the electrodes used and the method.

We supplemented the conclusion with the novelty of the method, as suggested by the reviewer:

“To the best of our knowledge, this is the first procedure using C-PB and BDDE for this purpose.”

Reviewer 3 Report

In my opinion the manuscript “Electrochemical Characterization and Voltammetric Determination of Methylisothiazolinone on a Boron-Doped Diamond Electrodeis suitable for publication in Molecules after minor revision. The presented manuscript is very interesting and topical. In my opinion, the work was written very carefully. Both the Introduction and the experimental part have been described in great details and properly interpreted, and the conclusions drawn are supported by appropriate experiments. I have some questions and suggestions that arose when I reviewed the submitted article. I hope they are useful for preparation of an improved version of the work: - Please describe in detail how the electrodes were prepared before the measurements and whether they were cleaned between measurements. - In Abstract and 2.4. Validation of the MIT determination method the linearity range and detection limit of MIT are given in mg/L while in other sections and in the descriptions of the figures the MIT concentration is given in mmol. I would like to ask you to unify the units throughout the work. - The effect of the buffer concentration on the MIT signal should be investigated. - The conclusions should state the advantage of the developed procedure for the determination of MIT compared to other voltammetric procedures for the determination of MIT.

Author Response

We would like to thank the Reviewer for their thoughtful review of the manuscript. It raise important issues and its inputs are very helpful for improving the manuscript. We agree with all comments and we have revised our manuscript accordingly.

We respond below in detail to each of the comments. In addition, we include how we have revised things. We hope that the reviewer will find our responses to their comments satisfactory.

Please, find below your comments repeated in italics and our responses inserted after each comment.

Yours sincerely,

Magdalena Jakubczyk

Response to comments from Reviewer

  1. Please describe in detail how the electrodes were prepared before the measurements and whether they were cleaned between measurements.

 The manuscript was supplemented with data on how to prepare the electrodes before and during the measurements:

“Prior to all the voltammetric experiments, the BDDE surface was mechanically polished with 0.01 μm alumina powder slurry on a polishing cloth, then sonicated in distilled water and dried. In the next stage electrode surface was cathodically activated in 1 mol L-1 H2SO4, at a potential of -2.4 V, for a period of 5 minutes [44,45]. This procedure was repeated each day before starting all voltammetric measurements. The BDDE surface was not require additional treatment between the registrations of the voltammetric curves.”

  1. In Abstract and 2.4. Validation of the MIT determination method the linearity range and detection limit of MIT are given in mg/L while in other sections and in the descriptions of the figures the MIT concentration is given in mmol. I would like to ask you to unify the units throughout the work.

The reviewer's comment has been taken into account.

  1. The effect of the buffer concentration on the MIT signal should be investigated.

It should be noted that the concentration of CP-B buffer components has no significant effect on the magnitude and shape of the MIT signal.

  1. The conclusions should state the advantage of the developed procedure for the determination of MIT compared to other voltammetric procedures for the determination of MIT.

We agree with the reviewer's remark. We made appropriate correction in the conclusion.

Round 2

Reviewer 2 Report

Thanks for the corrections and explanations. Manuscript can be accepted in this form.